# Strategies for high-temperature methyl iodide capture in azolate-based metal-organic frameworks

Tingting Pan[1,7], Kaijie Yang [1,7], Xinglong Dong [2,7], Shouwei Zuo[3], Cailing Chen [1], Guanxing Li[1], Abdul-Hamid Emwas[4], Huabin Zhang [3] & Yu Han [1,5,6] ✉

Efficiently capturing radioactive methyl iodide ($CH_3I$), present at low concentrations in the high-temperature off-gas of nuclear facilities, poses a significant challenge. Here we present two strategies for $CH_3I$ adsorption at elevated temperatures using a unified azolate-based metal-organic framework, MFU-4*l*. The primary strategy leverages counter anions in MFU-4*l* as nucleophiles, engaging in metathesis reactions with $CH_3I$. The results uncover a direct positive correlation between $CH_3I$ breakthrough uptakes and the nucleophilicity of the counter anions. Notably, the optimal variant featuring $SCN^-$ as the counter anion achieves a $CH_3I$ capacity of $0.41\,g\,g^{-1}$ at 150 °C under 0.01 bar, surpassing all previously reported adsorbents evaluated under identical conditions. Moreover, this capacity can be easily restored through ion exchange. The secondary strategy incorporates coordinatively unsaturated Cu(I) sites into MFU-4*l*, enabling non-dissociative chemisorption for $CH_3I$ at 150 °C. This modified adsorbent outperforms traditional materials and can be regenerated with polar organic solvents. Beyond achieving a high $CH_3I$ adsorption capacity, our study offers profound insights into $CH_3I$ capture strategies viable for practically relevant high-temperature scenarios.

As the electricity demand continues to rise and the need for a transition to a low-carbon economy becomes increasingly prominent, nuclear power is widely recognized as a promising alternative to traditional fossil fuels[1,2]. A significant challenge in the utilization of nuclear power involves the effective removal of volatile radionuclides from off-gas streams prior to their release into the atmosphere. Among these radionuclides, iodine isotopes, specifically [129]I and [131]I, constitute a critical component. These isotopes manifest as molecular iodine ($I_2$) and organic iodides, primarily methyl iodide ($CH_3I$)[3]. The adsorptive capture of $CH_3I$ is more difficult compared to

$I_2$, due to its lower concentration and the absence of strong intermolecular interactions, thus rendering it more susceptible to environmental dispersal[4,5]. While certain materials demonstrate high $CH_3I$ adsorption at ambient temperatures[6,7], the practical demands of off-gas treatment necessitate adsorbents capable of capturing low-concentration $CH_3I$ under elevated temperatures (~ 150 °C). The development of efficient adsorbents for high-temperature capture of $CH_3I$ remains a pressing yet challenging research objective. Furthermore, although irreversible capture is preferred in extreme situations like nuclear accidents to prevent the re-emission of

[1]Advanced Membranes and Porous Materials Center (AMPM), Physical Sciences and Engineering Division, King Abdullah University of Science and Technology (KAUST), Jeddah, Thuwal, Saudi Arabia. [2]School of Chemistry, University of Lincoln, Brayford Pool, Lincoln, United Kingdom. [3]KAUST Catalysis Center (KCC), Physical Sciences and Engineering Division, King Abdullah University of Science and Technology (KAUST), Jeddah, Thuwal, Saudi Arabia. [4]Imaging and Characterization Core Lab, King Abdullah University of Science and Technology (KAUST), Jeddah, Thuwal, Saudi Arabia. [5]School of Emergent Soft Matter, South China University of Technology, Guangzhou, China. [6]Center for Electron Microscopy, South China University of Technology, Guangzhou, China. [7]These authors contributed equally: Tingting Pan, Kaijie Yang, Xinglong Dong. ✉e-mail: hanyu@scut.edu.cn

radioactive materials[8], adsorbents that are easily regenerable hold significant value in most other scenarios.

Currently, adsorbents designed for high-temperature capture of CH₃I predominantly feature either amine functional groups or silver (Ag) particles. The lone pairs on the basic nitrogen (N) atoms in amine groups facilitate CH₃I binding through nucleophilic substitution reactions, leading to the formation of quaternary ammonium salts (Fig. 1a-i)[6,7,9–11]. However, this process is irreversible, posing challenges for adsorbent regeneration and capacity recovery. The Ag-based adsorbents, primarily Ag supported on zeolites, can immobilize CH₃I through dissociative chemisorption, forming silver iodide (AgI)[12–16]. The adsorption capacity of Ag zeolites for CH₃I mainly depends on the quantity of dispersed Ag species (e.g., Ag⁺ ions or charged/metallic clusters), which, in turn, is governed by the zeolite's cation exchange capacity (CEC). When the quantity of introduced Ag surpasses the CEC, Ag particles begin to develop. These particles exhibit a limited CH₃I adsorption capacity as only surface atoms are involved in the dissociative chemisorption process (Fig. 1a-ii)[17–19]. Moreover, the regeneration of Ag generally requires high-temperature hydrogen treatment[20]. Coordinatively unsaturated metal sites, hydrogen bonding, and halogen bonding have also been employed to promote CH₃I capture[10,21,22]. However, the inherent weakness of these interactions results in either limited adsorption capacity or facile desorption of CH₃I under high-temperature conditions.

In this study, we develop two strategies for the adsorptive capture of CH₃I at high temperatures, based on a single material platform: an azolate-based metal-organic framework (MOF), specifically MFU-4*l* (Zn₅Cl₄(BTDD)₃, where BTDD²⁻ = bis(1,2,3-triazolo[4,5-*b*],[4′,5′-*i*]) dibenzo[1,4]dioxin)). The intrinsic microporosity and excellent thermal stability of MFU-4*l* position it as a promising candidate for high-temperature CH₃I adsorption applications. As illustrated in Fig. 1c, the "Kuratowski-type" secondary building unit of MFU-4*l* features a centrally located, octahedrally coordinated Zn atom, accompanied by four peripheral, tetrahedrally coordinated Zn atoms, each bonded to a counter anion as a terminal ligand[23–25]. Intriguingly, both the tetra-coordinated Zn centers and counter anions in MFU-4*l* are exchangeable, affording tunable functionality and facile reversibility.

The first strategy employs counter anions in MFU-4*l* to serve as nucleophiles for the metathesis reaction with CH₃I (Fig. 1b-i). Unlike previous approaches that utilized embedded amine groups, these counter anions are easily removable post-methylation. Moreover, the generated coordinating I⁻ can be replaced by fresh counter anions via ion exchange, thus enabling full regeneration of the adsorbent material. To validate this strategy, the original MFU-4*l* variant, designated as MFU-Zn-Cl (with Cl⁻ as the counter anion), is transformed into MFU-Zn-X (where X = OH or SCN) via ion exchange. Assessment of these MFU-4*l* variants for CH₃I capture under various conditions indicates that their adsorption capacities positively correlate with the nucleophilicity of the counter anion, following the order: SCN⁻ > OH⁻ > Cl⁻.

The second strategy exploits the exchangeable metal sites in MFU-4*l* to enhance CH₃I adsorption via non-dissociative chemisorption (Fig. 1b-ii). Specifically, coordinatively unsaturated Cu(I) sites are incorporated into MFU-4*l* by exchanging tetra-coordinated Zn(II) with Cu(II) followed by reduction. These Cu(I) sites interact with the iodine end of CH₃I, thereby facilitating high-temperature CH₃I uptake. Notably, while Cu(II) also exhibits a strong affinity for iodide[26], it is prone to

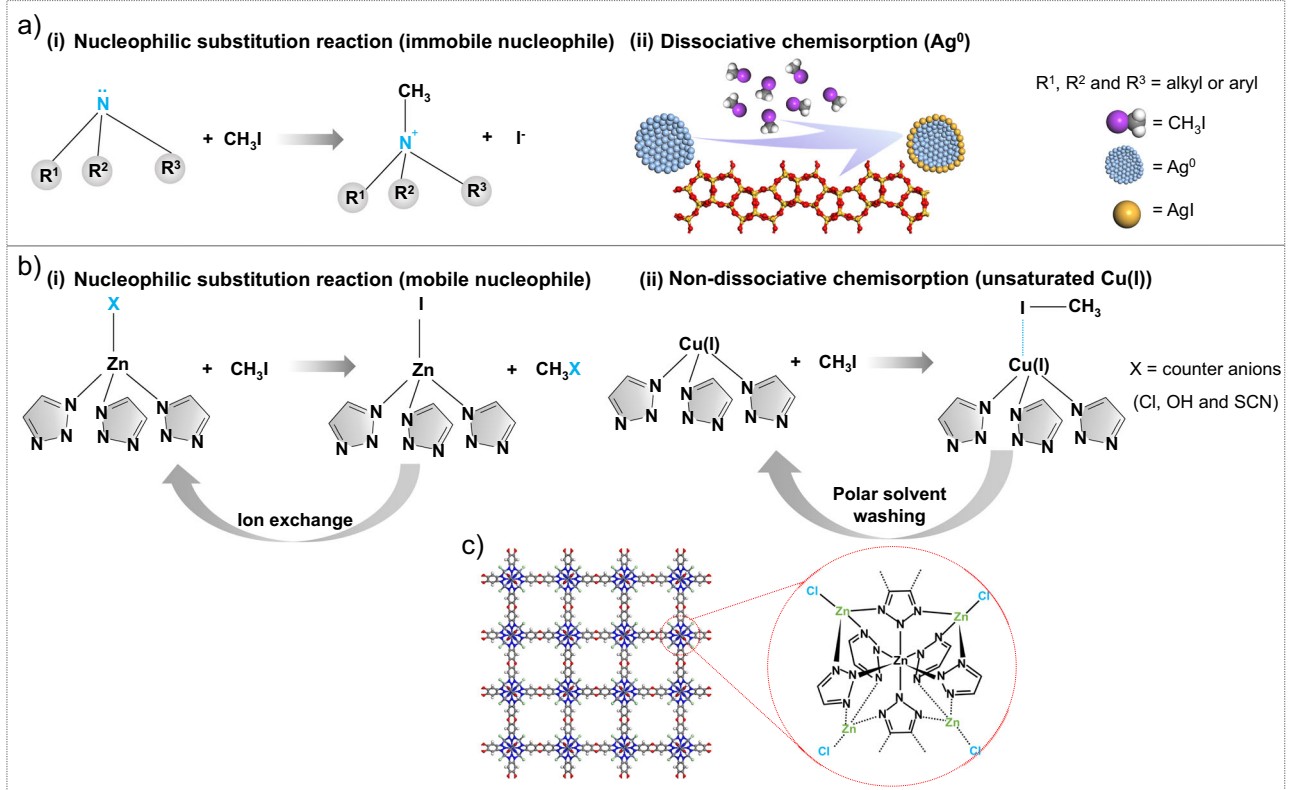

**Fig. 1 | Comparison of traditional and proposed methyl iodide adsorption mechanisms. a** Illustration of traditional mechanisms for the adsorptive capture of CH₃I: (i) the nucleophilic substitution reaction utilizing embedded amine groups as nucleophiles, and (ii) dissociative chemisorption whereby supported Ag clusters/particles react with CH₃I to form AgI. **b** Illustration of two mechanisms proposed in this study for the adsorptive capture of CH₃I: (i) a nucleophilic substitution reaction utilizing mobile nucleophiles, specifically the counter anions of MOF MFU-4*l*, and (ii) non-dissociative chemisorption through coordinatively unsaturated copper (I) sites within MFU-4*l*. Both strategies enable enhanced CH₃I uptake and easy adsorbent regeneration. **c** Structural representation of MFU-4*l*, with an enlarged view of the node structure showcasing the octahedrally coordinated Zn, tetrahedrally coordinated Zn, and counter anion Cl⁻. In the sphere-rod model, the color codes are Gray for C; blue for N; red for O; white for H; violet-gray for Zn; and green for Cl.

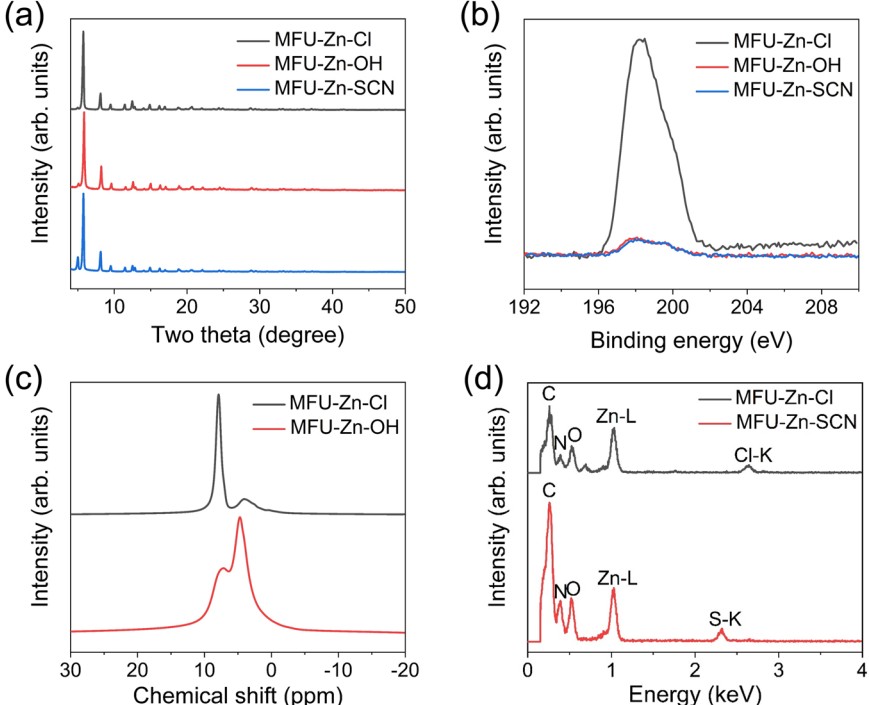

**Fig. 2 | Characterization results of MFU-4*l* variants, MFU-Zn-X (X = Cl, OH, and SCN). a** PXRD patterns and **b** Cl 2*p* XPS spectra of all three variants. **c** Solid-state ¹H NMR spectra of MFU-Zn-Cl and MFU-Zn-OH. **d** EDX spectra of MFU-Zn-Cl and MFU-Zn-SCN. Peaks corresponding to carbon (C), nitrogen (N), oxygen (O), L-edge of zinc (Zn), K-edge of chlorine (Cl), and sulfur (S) are labeled.

redox reactions with $CH_3I$, releasing undesirable $I_2$ as a byproduct[19]. The use of open Cu(I) sites circumvents this issue while maintaining high $CH_3I$ adsorption capacity. The MFU-4*l* variant with open Cu(I) sites, termed MFU-Cu(I), proves to be especially effective for capturing $CH_3I$ under low-concentration conditions. The adsorbed $CH_3I$ can be easily collected by washing with a polar organic solvent to regenerate the adsorption capacity.

The present study not only introduces two innovative and efficient strategies for high-temperature $CH_3I$ adsorption but also provides insights into host-guest interactions involving $CH_3I$. These contributions are pivotal to the design of adsorbents for practical nuclear off-gas treatment applications.

## Results

### Metathesis reaction using counter anions as nucleophiles

We synthesized MFU-Zn-Cl using a previously reported method[23]. To explore the effect of counter anions on $CH_3I$ capture, we derived two additional MFU-4*l* variants, MFU-Zn-OH and MFU-Zn-SCN, via anion exchange starting from MFU-Zn-Cl. Detailed synthetic procedures can be found in the Methods section. The rationale behind selecting Cl⁻, OH⁻, and SCN⁻ as counter ions lies in their distinct nucleophilicities. Nucleophilicity quantifies a nucleophile's ability to react at an electron-deficient center, which, within the context of $CH_3I$ adsorption, is the $CH_3$ group (Fig. 1b), under kinetically controlled conditions. Using the Swain-Scott equation, the nucleophilicities of various nucleophiles have been measured based on their reactivities relative to water using methyl bromide as the substrate[27]. The results at 25 °C indicate the sequencing of the nucleophilicities of the three anions as Cl⁻ (2.7) < OH⁻ (4.2) < SCN⁻ (4.4). It should be noted that a minimal difference in nucleophilicity values corresponds to a significant difference in reactivity, attributable to their logarithmic relationship.

The three materials displayed nearly identical powder X-ray diffraction patterns, with slight variations observed in the relative intensity of reflections, indicating that the crystallinity of MFU-4*l* was well preserved during the anion exchange processes (Fig. 2a). The X-ray photoelectron spectroscopy revealed negligible presence of Cl⁻ at ~198.3 eV in both MFU-Zn-OH and MFU-Zn-SCN[28], indicating a substantial degree of anion exchange (Fig. 2b). The successful incorporation of OH⁻ into MFU-4*l* was confirmed through solid-state ¹H nuclear magnetic resonance (NMR) spectroscopy, as evidenced by the emergence of a peak at ~4.7 ppm attributed to the protons of hydroxyl groups (Fig. 2c)[29]. Moreover, the incorporation of SCN⁻ was supported by the detection of an S signal in the energy-dispersive X-ray spectrum (Fig. 2d). Assuming complete anion exchange, the content of counter anions in the three samples was determined based on the Zn content measured from inductively coupled plasma-optical emission spectroscopy: OH⁻ (3.34 mmol g⁻¹) > Cl⁻ (3.25 mmol g⁻¹) > SCN⁻ (3.06 mmol g⁻¹) (Supplementary Table 1). Thermogravimetric analyses revealed that all three samples are thermally stable with the onset decomposition temperatures exceeding 400 °C (Supplementary Fig. 1).

Despite the broadly acknowledged significance, research exploring $CH_3I$ capture under dynamic, high-temperature conditions is scarce. The only two studies evaluating adsorbents at high temperatures (150 °C) employed a $CH_3I$ partial pressure of 0.2 bar (200,000 ppmv)[11,21], which is substantially higher than the $CH_3I$ concentration in practical off-gases. In the current study, an initial assessment was conducted on the dynamic $CH_3I$ capture performance of MFU-4*l* variants at 150 °C with 0.2 bar $CH_3I$ to directly compare with previously reported adsorbents. Following this, their adsorption capabilities were examined at a reduced $CH_3I$ concentration of 0.01 bar (10,000 ppmv), offering greater relevance to practical applications. The dynamic $CH_3I$ capture experiments were performed using a column breakthrough setup, as described in the Methods section. While the breakthrough curves generally exhibit a gradual elevation to reach a plateau, the retention times corresponding to initial breakthrough points were meticulously chosen for the calculation of adsorption capacities to preclude any $CH_3I$ leakage.

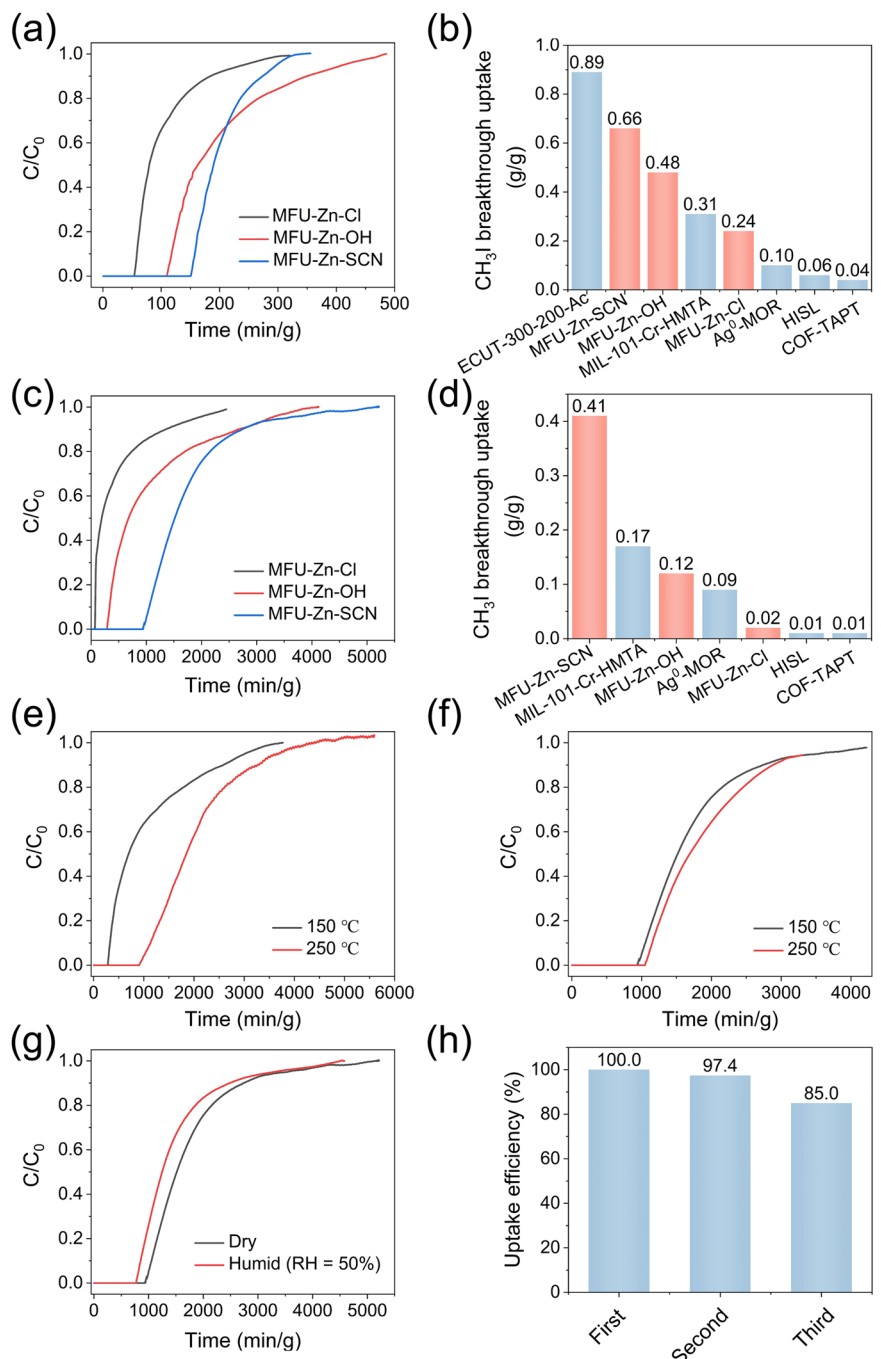

**Fig. 3 | Dynamic CH₃I adsorption performances of MFU-4*l* variants, MFU-Zn-X (X = Cl, OH, and SCN).** **a** CH₃I breakthrough curves obtained at 150 °C and a CH₃I partial pressure of 0.2 bar. **b** Comparison of CH₃I breakthrough uptakes across different adsorbents. Note that all uptake values stem from breakthrough curves collected using the same setup under conditions described in (**a**), with the exception of the data for ECUT-300-200-Ac, which was sourced from the literature. **c** CH₃I breakthrough curves obtained at 150 °C and a CH₃I partial pressure of 0.01 bar. **d** Comparison of CH₃I breakthrough uptakes across different adsorbents.

Note that all uptake values stem from breakthrough curves collected using the same setup under conditions described in (**c**). **e**, **f** CH₃I breakthrough curves for (**e**) MFU-Zn-OH and (**f**) MFU-Zn-SCN at 150 and 250 °C, under a CH₃I partial pressure of 0.01 bar. **g** CH₃I breakthrough curves for MFU-Zn-SCN, obtained at 150 °C with a CH₃I partial pressure of 0.01 bar, under dry and humid (RH = 50%) conditions. **h** Results from the recyclability tests for CH₃I adsorption using MFU-Zn-SCN at 150 °C under a CH₃I partial pressure of 0.01 bar.

Figure 3a shows the breakthrough profiles of MFU-Zn-X (X = Cl, OH, and SCN) obtained at 150 °C and the CH₃I partial pressure of 0.2 bar. It is interesting to note that despite possessing identical crystal structures and comparable anion contents, the three adsorbents manifested divergent CH₃I uptake capacities under identical test conditions. Specifically, their CH₃I uptake capacities determined based on the retention time of CH₃I breakthrough points follow the order:

MFU-Zn-Cl (0.24 g g⁻¹) < MFU-Zn-OH (0.48 g g⁻¹) < MFU-Zn-SCN (0.66 g g⁻¹) (Fig. 3b), which aligns consistently with the nucleophilicity of the three respective anions. However, the CH₃I adsorption capacities measured at room temperature did not mirror this trend, instead displaying minimal differences (see supplementary Fig. 2). These findings strongly suggest that at elevated temperatures, CH₃I uptake is predominantly governed by the binding strength that is associated

with the nature of the adsorptive sites. For MFU-4$l$ variants, the nucleophilicity of the counter anion emerges as a crucial determinant of high-temperature $CH_3I$ capture performance. Enhanced nucleophilicity leads to accelerated reaction kinetics, thereby boosting dynamic uptake capacity.

We synthesized a series of adsorbents previously developed for $CH_3I$ capture: all-silica zeolite HISL, nitrogen-rich covalent organic framework COF-TAPT, amine-modified metal-organic framework MIL-101-Cr-HMTA, and Ag-containing zeolite Ag$^0$-MOR, using reported methods. Their $CH_3I$ uptake capacities were assessed under consistent conditions (150 °C and 0.2 bar $CH_3I$), alongside the three MFU-4$l$ variants (Fig. 3b and Supplementary Fig. 3). COF-TAPT, HISL, and Ag$^0$-MOR showed low $CH_3I$ uptake ($\leq 0.1$ g g$^{-1}$) under the employed conditions, due to the lack or limited number of strong adsorptive sites, despite their effective $CH_3I$ adsorption at room temperature as previously reported. MIL-101-Cr-HMTA exhibited a high $CH_3I$ uptake of 0.31 g g$^{-1}$ under the same conditions, attributable to its tertiary amine groups which interact with $CH_3I$ through a nucleophilic substitution reaction (Fig. 1a-i). However, its $CH_3I$ uptake remains inferior to those of MFU-Zn-OH and MFU-Zn-SCN, possibly due to the lower reactivity of its nucleophilic sites. In summary, among all the adsorbents assessed, MFU-Zn-SCN demonstrated the highest $CH_3I$ uptake. It is worth noting that an exceptional $CH_3I$ uptake of 0.89 g g$^{-1}$ under similar conditions was reported for a bimetallic mesoporous MOF adsorbent ECUT-300-200-Ac (metallic centers: Cd and U) (Fig. 3b). However, our attempts to synthesize this material, following the reported method, were unsuccessful, precluding independent evaluation of its performance.

Figure 3c presents the breakthrough profiles of the MFU-4$l$ variants obtained at 150 °C with a $CH_3I$ partial pressure of 0.01 bar. Despite a reduction in $CH_3I$ uptake across all three materials in response to decreased $CH_3I$ concentration, a consistent trend was observed: MFU-Zn-Cl (0.02 g g$^{-1}$) < MFU-Zn-OH (0.12 g g$^{-1}$) < MFU-Zn-SCN (0.41 g g$^{-1}$). These findings further corroborate the established correlation between the nucleophilicity of counter anions and the high-temperature dynamic $CH_3I$ uptake capacity of MFU-4$l$. As depicted in Fig. 3d and Supplementary Fig. 4, MFU-Zn-SCN markedly outperforms other evaluated adsorbents. Notably, its $CH_3I$ uptake of 0.41 g g$^{-1}$ closely aligns with the theoretical value (0.43 g g$^{-1}$) calculated based on a one-to-one correspondence between the counter anion and the $CH_3I$ molecule. In contrast, all other tested adsorbents, except Ag$^0$-MOR, exhibit significantly lower $CH_3I$ uptakes than their theoretical values, which are derived from the quantities of their functional groups considered as potential adsorptive sites (Supplementary Table 1 and Supplementary Fig. 5). This finding indicates that under high-temperature, low-concentration conditions, the equilibrium adsorption capacity is exclusively determined by the number of strong chemisorption sites, while high reactivity of the adsorptive site is essential for achieving this capacity. The extraordinary $CH_3I$ capture performance of MFU-Zn-SCN can thus be ascribed to the presence of abundant, highly nucleophilic counter anions.

To confirm that the $CH_3I$ adsorption by MFU-4$l$ variants proceeds through the proposed mechanism (Fig. 1b-i), we carefully characterized the samples before and after the adsorption of $CH_3I$. The significant reduction in the powder X-ray diffraction (PXRD) peak intensity upon $CH_3I$ adsorption indicates the incorporation of a large amount of guest molecules into the channels (Supplementary Fig. 6). The effective adsorption of $CH_3I$ within the three MFU-4$l$ variants is further corroborated by results from energy-dispersive X-ray (EDX) spectroscopy elemental mapping, displaying a uniform distribution of iodine in the samples post-adsorption. Concurrently, the EDX reveals a significant decrease in the content of Cl in MFU-Zn-Cl, O in MFU-Zn-OH, and S in MFU-Zn-SCN upon $CH_3I$ adsorption (Fig. 4d–f). The disappearance of the characteristic band at 2062 cm$^{-1}$ in the Fourier-transform infrared spectroscopy affirms the elimination of SCN$^-$ from

MFU-Zn-SCN during $CH_3I$ adsorption (Supplementary Fig. 7). Moreover, elemental analysis reveals a marked reduction in the O/Zn molar ratio in MFU-Zn-OH from 4.45 to 3.26 during $CH_3I$ adsorption (Supplementary Table 2). These results unequivocally demonstrate the substitution of counter anions by I$^-$ in the three MFU-4$l$ variants. In addition, real-time mass spectrometry analysis of the effluent gas demonstrates the formation of $CH_3Cl$, $CH_3OH$, and $CH_3SCN$ from MFU-Zn-Cl, MFU-Zn-OH, and MFU-Zn-SCN, respectively, upon $CH_3I$ adsorption (Fig. 4a–c).

Collectively, the aforementioned characterization results substantiate that $CH_3I$ is captured by the adsorbents via a metathesis reaction with counter anions of MFU-4$l$ (Fig. 1b-i). The reaction essentially constitutes a nucleophilic substitution, wherein the counter anion acts as a nucleophile to attack the carbon end of $CH_3I$ and replace I$^-$, while the liberated I$^-$ coordinates with Zn$^{2+}$ within the MFU-4$l$ framework. The generated volatile compounds $CH_3Cl$, $CH_3OH$, and $CH_3SCN$ are expelled from the adsorbent concomitantly with the carrier gas.

For conventional physical adsorption processes, an inverse relationship generally exists between adsorption capacity and temperature. Consequently, efficient adsorptive capture of $CH_3I$ at elevated off-gas temperatures (-150 °C) is unattainable via physical adsorption, necessitating the employment of chemical adsorption mechanisms. Should the MFU-4$l$ variants indeed capture $CH_3I$ through the nucleophilic substitution reaction, wherein the nucleophilicity of the counter anion affects the reaction kinetics and, consequently, the breakthrough uptake, an augmentation in $CH_3I$ uptake could be anticipated with a rise in temperature-enhancing reaction rate. To verify this hypothesis, we further evaluated MFU-Zn-OH and MFU-Zn-SCN at an elevated temperature of 250 °C, while keeping other conditions unchanged. Interestingly, MFU-Zn-OH exhibited a more than threefold enhancement in $CH_3I$ uptake at 250 °C compared to 150 °C (0.39 vs. 0.12 g g$^{-1}$) (Fig. 3e). Conversely, MFU-Zn-SCN exhibited a marginal increase in $CH_3I$ uptake from 0.41 to 0.45 g g$^{-1}$ with the same temperature alteration (Fig. 3f). These findings confirm that an increase in temperature can promote the dynamic $CH_3I$ capture based on chemical adsorption, particularly when the reaction rate is the limiting factor.

Considering the consistent presence of moisture with iodine species in nuclear off-gas, the optimal adsorbent, MFU-Zn-SCN, was further tested at 150 °C and 0.01 bar $CH_3I$ with 50% relative humidity. Results showed a moderate decline in $CH_3I$ uptake to 0.33 g g$^{-1}$ in humid conditions (Fig. 3g). Nonetheless, MFU-Zn-SCN's $CH_3I$ uptake under these conditions remains significantly higher than that of other adsorbents tested under dry conditions (Fig. 3d).

A notable benefit of using MFU-4$l$ variants for $CH_3I$ capture lies in their easy regeneration post-adsorption through simple anion exchange. For example, to restore the adsorption capacity of spent MFU-Zn-SCN, the sample was extensively rinsed with a 0.2 M lithium thiocyanate aqueous solution. The effective iodide release was confirmed by UV/Vis spectroscopy and EDX analysis (Supplementary Fig. 8a, b); the regenerated MFU-Zn-SCN displayed similar PXRD spectra and FTIR spectrum as the pristine material, attesting to the efficacy of the regeneration process (Supplementary Fig. 8c, d). Over three consecutive adsorption/regeneration cycles, the $CH_3I$ uptake of MFU-Zn-SCN consistently retained above 85% of its initial capacity (Fig. 3h).

## Non-dissociative chemisorption on Cu(I) sites

In addition to counter anions, the metal centers within MFU-4$l$ can also be utilized to achieve efficient $CH_3I$ capture. To this end, MFU-Zn-Cl was first subjected to cation exchange with $CuCl_2$; the resultant green powder, named MFU-Cu(II)-Cl, was then converted to beige-colored MFU-Cu(I) through anion exchange with lithium formate (HCOOLi) followed by vacuum heating treatment (see Fig. 5a). This method

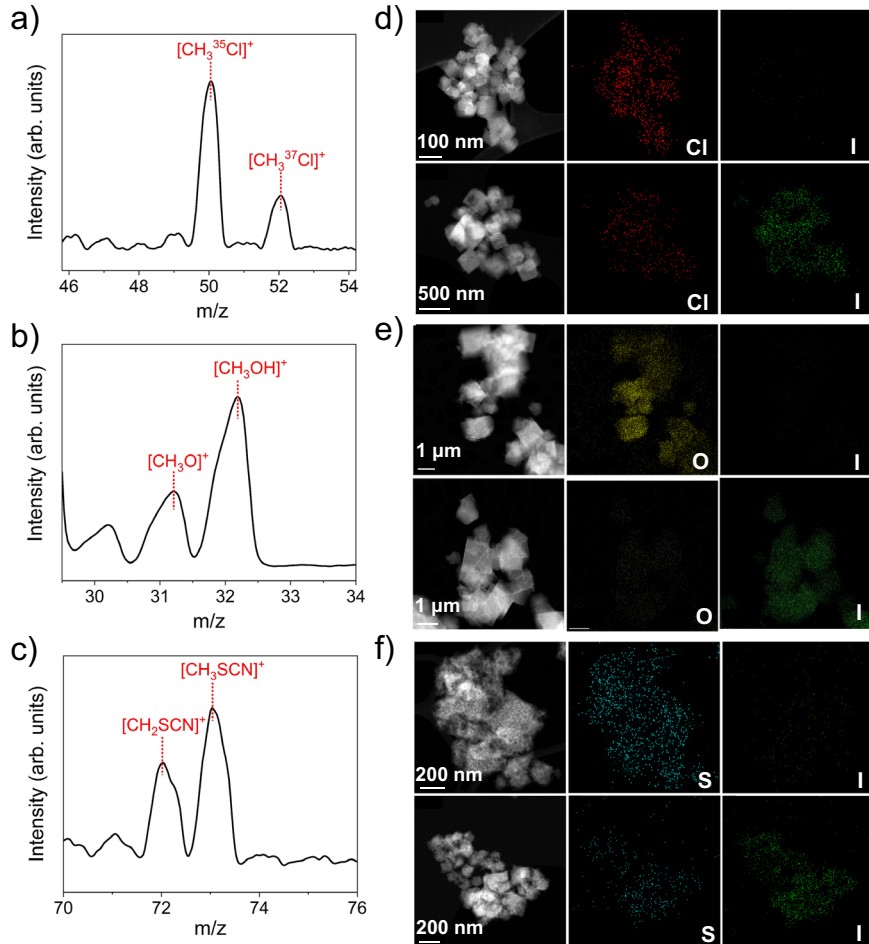

**Fig. 4 | Characterization of CH₃I/MFU-Zn-X (X = Cl, OH, and SCN) interactions.**
**a**–**c** Mass spectra of the effluent gas from the adsorption column for MFU-Zn-Cl (**a**), MFU-Zn-OH (**b**), and MFU-Zn-SCN (**c**). **d**–**f** High-angle annular dark-field scanning transmission electron microscopy images paired with their corresponding EDX elemental maps for MFU-Zn-Cl (**d**), MFU-Zn-OH (**e**), and MFU-Zn-SCN (**f**). For each panel, the top row shows results prior to CH₃I adsorption, while the bottom row shows results post-adsorption.

generates coordinatively unsaturated Cu(I) sites in MFU-4*l*, rendering unique adsorption and separation capabilities for $H_2$, $N_2O$, $O_2$, and $N_2$[30–33]. However, the substantial potential of these Cu(I) sites for $CH_3I$ capture remains unexplored.

In MFU-Cu(II)-Cl and MFU-Cu(I), the Zn/Cu molar ratio is about 2.65/2.35 as determined by inductively coupled plasma-optical emission spectrometry (ICP-OES), indicating ~60% of tetrahedrally coordinated Zn has been replaced by Cu. The transformation from divalent to monovalent Cu during the vacuum heating process, with formate as a reducing agent, is corroborated by Cu 2*p* X-ray photoelectron spectroscopy (XPS) characterization. The XPS spectrum of MFU-Cu(II)-Cl displays two satellite peaks at 942.3 and 962.4 eV, alongside peaks at 934.6 and 954.5 eV, corresponding to Cu 2$p_{3/2}$ and 2$p_{1/2}$ of divalent Cu. In contrast, the XPS spectrum of MFU-Cu(I) exhibits two main peaks at 932.4 and 952.3 eV, attributed to Cu 2$p_{3/2}$ and 2$p_{1/2}$ of monovalent Cu (Supplementary Fig. 9a). These observations are consistent with the results reported in the literature[34]. Further confirmation of the lower oxidation state in MFU-Cu(I) compared to MFU-Cu(II)-Cl, which exhibits characteristic $Cu^{2+}$ features, is provided by electron paramagnetic resonance (EPR) and X-ray absorption near edge structure (XANES) spectroscopy results (Supplementary Fig. 9b, c). Substituting divalent Zn in MFU-4*l* with monovalent Cu results in the latter being coordinatively unsaturated, a necessary condition for maintaining charge balance. The resultant coordinatively unsaturated Cu(I) sites are anticipated to be capable

of capturing $CH_3I$ through non-dissociative chemical adsorption. In the $k^2$-weighted extended X-ray absorption fine structure (EXAFS), MFU-Cu(I) features a main peak at ~1.48 Å, corresponding to the first coordination shell of Cu-N. The fitting result reveals a coordination number of 3.7, exceeding the anticipated value of 3. This discrepancy may arise as certain open Cu sites were likely terminated due to inadvertent air exposure during measurement (Fig. 5f, Supplementary Table 3, and Supplementary Fig. 10a).

We evaluated the $CH_3I$ capture ability of MFU-Cu(I) at 150 °C using the same column breakthrough setup as described in the previous section. At a $CH_3I$ partial pressure of 0.2 bar, MFU-Cu(I) showed over double the $CH_3I$ uptake of the original MFU-Zn-Cl material (0.51 vs. 0.24 g g⁻¹). This value is comparable to that of MFU-Zn-OH, slightly lower than that of MFU-Zn-SCN, while markedly higher than most previously reported adsorbents (Supplementary Fig. 11a, b). When the $CH_3I$ pressure was lowered to 0.01 bar, the difference in $CH_3I$ uptake between MFU-Cu(I) and MFU-Zn-Cl became more significant (0.14 vs. 0.02 g g⁻¹). These results clearly demonstrate the pivotal role of coordinatively unsaturated Cu(I) sites in promoting $CH_3I$ adsorption at high-temperature, low-concentration conditions. Although the use of lithium formate during the preparation of MFU-Cu(I) may lead to the presence of residual formate as counter anions, their influence on $CH_3I$ uptake is expected to be marginal because of their small amount and weak nucleophilicity. Under the extreme conditions employed (150 °C and 0.01 bar $CH_3I$), MFU-Cu(I) ranks third in $CH_3I$ uptake capacity

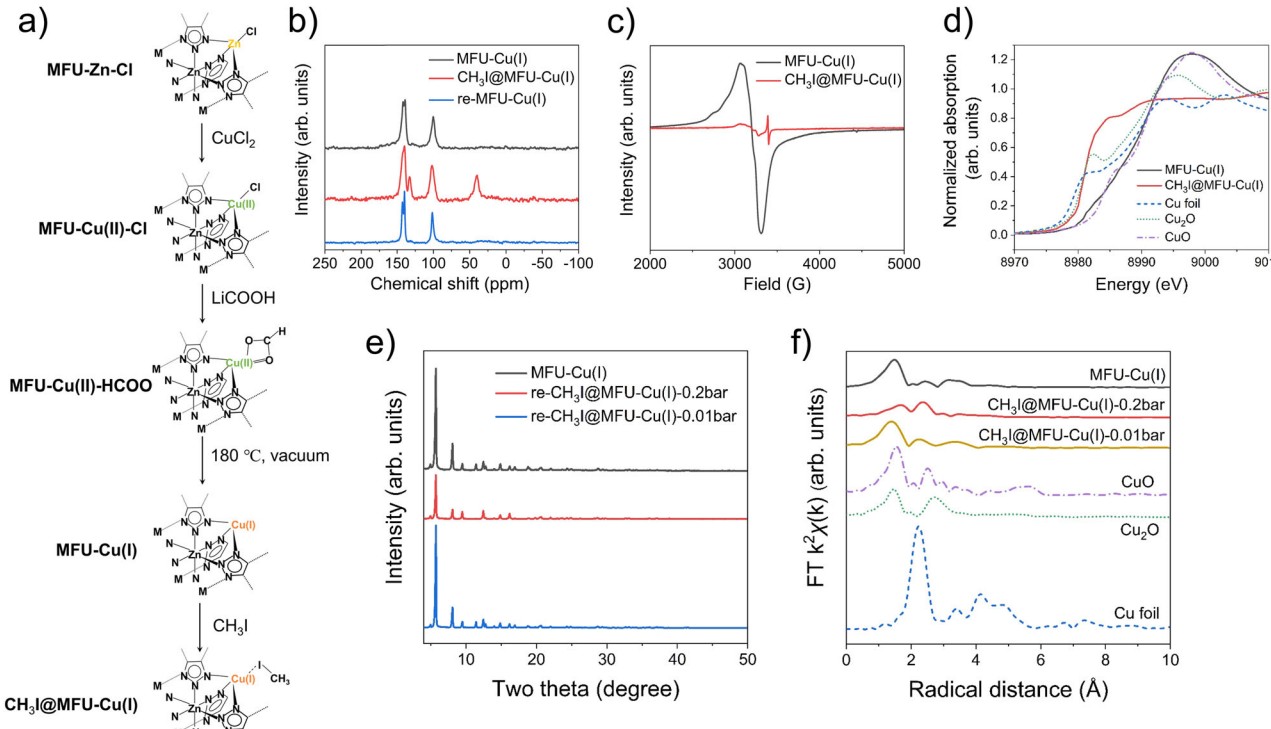

**Fig. 5 | Characterization of CH₃I/MFU-Cu(I) interactions. a** Illustration of the synthetic route from MFU-Zn-Cl to MFU-Cu(I) and its subsequent CH₃I adsorption. Note that the interaction mode between CH₃I and MFU-Cu(I) was obtained through DFT structure optimization, as shown in Supplementary Fig.12. **b** Solid-state ¹³C NMR spectra of pristine MFU-Cu(I), CH₃I-adsorbed MFU-Cu(I) (designated as CH₃I@MFU-Cu(I)), and the regenerated MFU-Cu(I) (designated as re-MFU-Cu(I)). **c** EPR spectra of MFU-Cu(I) and CH₃I@MFU-Cu(I). **d** Cu K-edge X-ray absorption near edge structure spectra of MFU-Cu(I) and CH₃I@MFU-Cu(I), along with Cu foil, Cu₂O, and CuO as reference materials. **e** PXRD patterns of pristine MFU-Cu(I) and two regenerated MFU-Cu(I) samples, which underwent CH₃I adsorption at 0.2 bar and 0.01 bar of CH₃I, respectively. **f** Fourier-transformed k²-weighted EXAFS spectra of pristine MFU-Cu(I) and two MFU-Cu(I) samples subjected to CH₃I adsorption at pressures of 0.2 bar and 0.01 bar, respectively. Included for comparison are the spectra of reference materials: Cu foil, Cu₂O, and CuO.

among all evaluated adsorbents, surpassed only by MFU-Zn-SCN and MIL-101-Cr-HMTA (Supplementary Fig. 11c, d).

A variety of characterizations were performed on MFU-Cu(I) before and after CH₃I adsorption. Prior to these characterizations, the adsorbent was subjected to degassing at 150 °C under vacuum for 12 h to remove the physically adsorbed CH₃I. In the solid-state ¹³C NMR spectra, a new peak at ~40 ppm emerged upon adsorption, indicative of the carbon atom in CH₃I (Fig. 5b). This evidence suggests the retention of CH₃I as an intact molecule, affirming its capture via non-dissociative chemisorption. EPR spectroscopy revealed that the Cu species in MFU-Cu(I) stably retains its monovalent state post CH₃I adsorption (Fig. 5c). Normalized Cu K-edge XANES spectra show that the CH₃I adsorption induces a slight shift of the absorption edge to a lower energy, approximating that of Cu₂O (Fig. 5d). These findings imply that the intact CH₃I molecule is bonded to the adsorbent through the interaction of its electron-rich moiety, specifically the iodine end, with the open Cu(I) site (Fig. 5a). Theoretical calculations indicate a horizontal configuration of CH₃I on the Cu(I) site (Supplementary Fig. 12).

The non-dissociative chemisorption of CH₃I on the open Cu(I) sites enables its facile desorption via washing with a strong polar solvent (DMF followed by methanol), as evidenced by the elimination of the C peak corresponding to CH₃I in the solid-state ¹³C NMR spectra (Fig. 5b). This fact suggests the potential for easy regeneration of MFU-Cu(I) for successive CH₃I capture cycles. However, it is observed that the MFU-Cu(I) subjected to a CH₃I capture test at 150 °C and 0.2 bar exhibits a significant reduction (~43.5 %) in uptake capacity in the subsequent test (Supplementary Fig. 13). This diminished capacity is ascribed to the degraded crystalline structure

of the regenerated MFU-Cu(I) relative to the fresh sample, as evidenced by PXRD (Fig. 5e). Compared to the fresh MFU-Cu(I), the post-adsorption MFU-Cu(I) shows an additional peak at ~2.35 Å in EXAFS, likely attributable to the Cu-I bond (Fig. 5f). The fitting of the spectrum indicates coordination numbers of 1.8 for Cu-I and 2.2 for Cu-N (Supplementary Table 3 and Supplementary Fig. 10b). This data suggests that, on average, each Cu center is coordinated with approximately two I and two N atoms, aligning with the measured CH₃I uptake capacity of 0.51 g g⁻¹. Assuming that high-temperature CH₃I adsorption occurs solely at strong chemical adsorption sites, specifically the open Cu(I) sites, with a one-to-one correspondence, the theoretical CH₃I uptake, calculated based on Cu content, is approximately 0.24 g g⁻¹. Hence, the structural degradation of MFU-Cu(I) results from the substitution of Cu-N by Cu-I, occurring under conditions of relatively high CH₃I concentration.

When tested at 150 °C and 0.01 bar, MFU-Cu(I) exhibits the ability to fully restore its CH₃I uptake capacity following regeneration through washing with polar solvents (Supplementary Fig. 13). This favorable outcome is attributed to the low CH₃I concentration under which each open Cu(I) site adsorbs less than one CH₃I molecule on average, as determined based on the measured CH₃I uptake (0.14 g g⁻¹) and corroborated by EXAFS analysis. The EXAFS spectrum of the MFU-Cu(I) sample after CH₃I adsorption at 0.01 bar resembles that of the pristine MFU-Cu(I) (Fig. 5f), with fitting results revealing coordination numbers of 0.8 for Cu-I and 3.1 for Cu-N (Supplementary Table 3). Thus, MFU-Cu(I) avoids structural degradation, a fact validated by PXRD (Fig. 5e). These findings lead to the conclusion that MFU-Cu(I) is particularly effective for CH₃I capture under low-concentration conditions.

## Discussion

In conclusion, we have developed two innovative strategies for high-temperature $CH_3I$ capture: the nucleophilic substitution reaction using mobile nucleophiles and non-dissociative chemisorption utilizing coordinatively unsaturated Cu(I) sites. The efficacy of both strategies has been showcased using the MOF MFU-4*l* as a singular material platform, leveraging the high tunability of its secondary building units. The prepared MFU-4*l* variants, each with different counter anions, exhibit varying $CH_3I$ uptake capacities at 150 °C during dynamic breakthrough tests. These capacities positively correlate with the nucleophilicity of the counter anion, following the order: $SCN^- >$ $OH^- >$ $Cl^-$. These findings align with our expectations, validating the proposed adsorption mechanism based on the nucleophilic substitution reaction. The optimal adsorbent, MFU-Zn-SCN, exhibits a high $CH_3I$ breakthrough uptake of $0.41\,g\,g^{-1}$ at 150 °C with the partial pressure of 0.01 bar. This performance significantly surpasses various benchmark adsorbents tested under identical conditions. Importantly, the mobility of the coordinating counter anions ensures that the $CH_3I$-saturated MFU-Zn-SCN possesses commendable recyclability and can be effortlessly regenerated through straightforward ion exchange. Cu(I) integrated into MFU-4*l* serves as an alternative adsorptive site, showcasing excellent $CH_3I$ capture capability due to its strong coordination interaction with the iodine end of $CH_3I$. The adsorbent containing Cu(I) sites, MFU-Cu(I), displays a breakthrough $CH_3I$ uptake of $0.14\,g\,g^{-1}$ at 150 °C under a $CH_3I$ partial pressure of 0.01 bar. Although this capacity is less than that of MFU-Zn-SCN, developed via the primary strategy, it still exceeds the performance of various benchmark adsorbents. $CH_3I$ adsorbed on Cu(I) sites can be efficiently removed using polar solvents, allowing for the restoration of MFU-Cu(I)'s adsorption capacity. Our in-depth study has shed light on host-guest interactions involving $CH_3I$, enriching the fundamental understanding of high-temperature $CH_3I$ adsorption and provide guidance for the rational design of advanced adsorbents for real nuclear off-gas treatment.

## Methods

### Materials and characterization

All the starting materials and reagents utilized in this study were procured from commercial sources and employed without further purification. Anhydrous *N,N*-dimethylformamide (DMF, 99.8%), anhydrous *N,N*-dimethylacetamide (DMA, 99.8%), anhydrous methanol (MeOH, ≥99.9%), anhydrous dichloromethane ($CH_2Cl_2$, ≥99.9%), anhydrous acetonitrile ($CH_3CN$, ≥99.9%), anhydrous tetrahydrofuran (THF, ≥99.9%), formic acid (HCOOH, 88%), zinc chloride ($ZnCl_2$, 97%), copper chloride ($CuCl_2$, 97%), lithium carbonate ($Li_2CO_3$, ≥99%), lithium chloride (LiCl, ≥99%), sodium bicarbonate ($NaHCO_3$, ≥99.7%), and lithium thiocyanate hydrate ($LiSCN·xH_2O$) were all acquired from Sigma-Aldrich. Bis(1H-1,2,3-triazolo[4,5-b],[4′,5′-i])dibenzo[1,4]dioxin (H$_2$BTDD, >97%) was sourced from EXTENSION China. The synthesis of HISL and Ag$^0$-MOR followed established protocols as outlined in the literatures[15,35]. MIL-101-Cr-HMTA and COF-TAPT were generously provided by Dr. Baiyan Li[11] and Dr. Yaqiang Xie[6], respectively.

PXRD patterns were acquired utilizing a Bruker D8 Advance instrument operating with a scanning speed of 8°/min, employing Cu Kα radiation (λ = 0.1542 nm) at a voltage of 40 kV and a current of 40 mA. Nitrogen sorption isotherms at 77 K were obtained using a Micromeritics ASAP 2420 instrument. ICP-OES measurements were conducted employing an Agilent 5110 spectrometer. Elemental analyses of oxygen were carried out utilizing a Flash 2000 CHNO/S element analyzer. Thermogravimetric analysis (TGA) was performed employing a TGA-Iris 209 thermogravimetric analyzer over a temperature range of 25 °C to 800 °C, with a heating rate of 5 °C/min. ATR-FTIR spectra were recorded utilizing a Nicolet iS50 spectrometer from Thermo Fisher Scientific, with data collection spanning the range of 525 to 4000 cm$^{-1}$. Solid-state $^{13}C$ NMR spectra were acquired using a Bruker Avance III WB-400 instrument. XPS measurements were

executed on a Kratos AXIS Supra system. EPR spectra were obtained at ambient temperature using a Bruker EMX-10/12 EPR spectrometer operating in the X-band frequency range, employing the following parameters: microwave frequency of 9.8 GHz, microwave power of 20 mW, modulation frequency of 100 kHz, and a 10 dB attenuator. X-ray absorption spectroscopy (XAS) measurements were performed at the 1W1B beamline of the Beijing Synchrotron Radiation Facility (BSRF). Scanning TEM (STEM) imaging and EDX analysis were conducted utilizing an FEI Titan microscope equipped with a probe Cs corrector, operating at 300 kV. UV-Vis spectra were recorded at a scanning rate of 600 nm/min on a Varian Cary 5000 spectrophotometer.

### Preparation of MFU-Zn-X (X = Cl, OH, and SCN)

MFU-4*l* (MFU-Zn-Cl) was synthesized following the procedure outlined in previous literature[30]. In brief, H$_2$BTDD (1.46 mmol, 0.4 g) was dissolved in 400 mL of DMF under stirring and subjected to heating at 140 °C for 30 min. Subsequently, $ZnCl_2$ (30.1 mmol, 4.095 g) was introduced into the cooled solution, and stirring continued until complete dissolution of $ZnCl_2$ was achieved. The resulting solution was then heated under reflux at 140 °C with continuous stirring for 18 h, followed by cooling to room temperature. The resulting precipitate was isolated via filtration, washed successively with DMF (3 × 40 mL), $CH_3OH$ (3 × 40 mL), and $CH_2Cl_2$ (3 × 40 mL), and subsequently dried at 80 °C under vacuum for 12 h to afford a pale-yellow microcrystalline powder.

MFU-Zn-OH was synthesized by suspending MFU-Zn-Cl (0.16 mmol, 0.2 g) in a 20 mL aqueous solution of 0.1 M $NaHCO_3$[36]. Following a 30-min incubation period, the resulting sample underwent centrifugation, and the soaking procedure was iterated five times to enhance the degree of exchange. The resulting solids were then washed successively with water (3 × 20 mL) and anhydrous THF (3 × 20 mL) to yield MFU-Zn-$HCO_3$. Subsequent conversion to MFU-Zn-OH was achieved by subjecting MFU-Zn-$HCO_3$ to vacuum heating at 100 °C for 24 h.

MFU-Zn-SCN was prepared by immersing MFU-Zn-Cl (0.16 mmol, 0.2 g) in a 20 mL aqueous solution containing 0.2 M $LiSCN·xH_2O$. After 30 min, the resulting mixture underwent centrifugation, and the soaking process was repeated five times to augment the exchange efficiency. The resulting solids were washed sequentially with water (3 × 20 mL) and anhydrous THF (3 × 20 mL) to afford MFU-Zn-SCN.

### Preparation of MFU-Cu(I)

Prior to the synthesis of MFU-Cu(I), a solution of HCOOLi was prepared as described in the ref. 32. Typically, HCOOH (100 mmol, 4.6 g) was combined with a suspension of $Li_2CO_3$ (55 mmol, 4.06 g) in $CH_3OH$, and the resulting mixture was stirred for 30 min at room temperature. Excess $Li_2CO_3$ was removed via filtration, yielding a 0.5 M methanolic solution of HCOOLi.

The synthesis of MFU-Cu(I) proceeded through three distinct steps[32]. In the initial step, termed metal exchange, $CuCl_2$ (6 mmol, 0.8067 g) was dissolved in DMA (30 mL), and MFU-4*l* (0.12 mmol, 0.15 g) was introduced into the solution. The mixture was then heated at 60 °C under N$_2$ flow for 16 h, resulting in the formation of a green precipitate. This precipitate was subsequently isolated by filtration, washed with DMF (3 × 40 mL) and $CH_3CN$ (3 × 40 mL), and dried at 80 °C under vacuum for 12 h to yield a red-brown microcrystalline powder, denoted as MFU-Cu(II)-Cl.

In the second step, designated anion exchange, the MFU-Cu(II)-Cl powder obtained from the previous step was stirred with a 0.2 M solution of HCOOLi in $CH_3OH$ (50 mL) for 30 min at room temperature. The resulting precipitate was collected, washed with $CH_3OH$ (3 × 40 mL) and $CH_2Cl_2$ (3 × 40 mL), and then dried at 80 °C under vacuum for 12 h, yielding a green microcrystalline powder designated as MFU-Cu(II)-HCOO.

Finally, the MFU-Cu(II)-HCOO powder underwent vacuum heating up to 180 °C, with a heating rate of 5 K min⁻¹, and maintained at this temperature for 1 h. The resulting gray powder obtained from this process is identified as MFU-Cu(I).

## Dynamic methyl iodide adsorption experiments

Dynamic adsorption experiments were executed utilizing a laboratory-scale fixed-bed reactor operating at a temperature of 150 °C. Specifically, 30 mg of adsorbent was loaded into a quartz column (4.6 mm I.D. × 200 mm) with silane-treated glass wool employed to occupy the interstitial void space. The column was subsequently activated at 150 °C for 6 h under a dry helium flow (10 mL/min). To evaluate performance under a partial pressure of 0.2 bar, a nitrogen carrier flow at a rate of 3 mL/min was directed through a $CH_3I$ vapor generator, subsequently passing through the adsorbent-packed column at a linear velocity of 18 cm/min, corresponding to a residence time of approximately 5 s. The concentration of $CH_3I$ in the effluent was continuously monitored using an online mass spectrometry (MS) system. The flow rate of $CH_3I$ under these conditions was determined to be approximately 4.408 mg/min by trial and error.

To assess performance under a partial pressure of 0.01 bar, a nitrogen carrier flow at a rate of 0.3 mL/min was directed through the $CH_3I$ vapor generator, subsequently combined with another dry nitrogen flow at a rate of 5.7 mL/min before passing through the adsorbent column. Under these conditions, the linear velocity was 36 cm/min, with a residence time of ~2.5 s. The flow rate of $CH_3I$ under this configuration was approximately 0.426 mg/min. Experiments conducted under humid conditions (RH = 50%) at 0.01 bar involved mixing a nitrogen flow passing through the $CH_3I$ vapor generator at a rate of 0.3 mL/min with a nitrogen flow passing through the water vapor generator at a rate of 3 mL/min, along with a dry nitrogen flow at a rate of 2.7 mL/min. The combined gas stream was then directed through the adsorbent column. Monitoring of $CH_3I$ in the effluent was performed using an online MS system.

The adsorption capacities at breakthrough ($Q_{breakthrough}$) were determined using the following equation:

$$Q_{breakthrough} = \frac{v \times t_{1\%}}{m} 1$$

Where $v$ represents the flow rate of $CH_3I$ (mg/min); $t_{1\%}$ (corresponding to $C/C_0 = 1\%$, min) denotes the breakthrough point when the concentration of $CH_3I$ in the effluent stream (C) reaches 1% of the initial concentration ($C_0$); $m$ signifies the weight of the adsorbent (mg).

## Material regeneration

The regeneration of MFU-Zn-SCN entails an ion exchange process. Initially, 100 mg of $CH_3I$-saturated MFU-Zn-SCN was immersed in a 20 mL aqueous solution containing 0.2 M LiSCN·xH₂O. After a 30-min incubation period, the sample underwent centrifugation, with the soaking procedure iterated several times to enhance the degree of exchange. The progress of the exchange process was monitored by measuring the concentration of I⁻ species in the filtrate using ultraviolet-visible spectroscopy (UV-Vis). Results indicated that five exchange cycles were sufficient to extract all extractable I⁻ species. The regenerated sample was subsequently washed with water (3 × 20 mL) and anhydrous THF (3 × 20 mL) before being dried under vacuum at 80 °C overnight, in preparation for the subsequent $CH_3I$ adsorption experiment at 150 °C.

The regeneration protocol for MFU-Cu(I) involved several steps: Initially, 100 mg of $CH_3I$-loaded MFU-Cu(I) was washed with 20 mL of DMF for 30 min. Subsequently, the sample was collected via centrifugation and subjected to an additional wash with a methanol solution. Finally, the regenerated sample was dried under vacuum at

80 °C overnight, prior to its utilization in subsequent cycles of $CH_3I$ adsorption experiments.

## Data availability

All data supporting the findings of this study are available within the article and the Supplementary information file, or available from the corresponding authors on request. Source data are provided with this paper.

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

## Acknowledgements
This research is supported by the baseline fund to Y.H. (BAS/1/1372-01-01) from King Abdullah University of Science and Technology.

## Author contributions
Y.H. and T.P. conceived and designed this project. T.P., K.Y. and X.D. carried out the synthesis, characterization, and iodine adsorption experiments. S.Z. and H.Z. assisted with the XAS analysis. C.C. and G.L. assisted with the EDX elemental mapping. A.-H.E. assisted with the solid NMR and EPR test. Y.H. and T.P. wrote the manuscript. All the authors discussed the results and commented on the manuscript.

## Competing interests
The authors declare no competing interests.
