## [Peer Review File · Nature Communications]

Strategies for High-temperature Methyl Iodide Capture in Azolate-based Metal-Organic FrameworksREVIEWER COMMENTS

Reviewer #1 (Remarks to the Author):

In this manuscript, the authors present two innovative sorption methods for chemical capture, both of which can be implemented using a single MOF-based material, MFU-4l. Impressively, these methods show remarkable efficacy in capturing CH₃I at 150 °C, a challenging temperature that is rarely addressed in existing literature. The manuscript successfully demonstrates the utilization of MFU-4l's exchangeable counter anions and cations through two distinct strategies, clearly elucidating the mechanism behind their exceptionally high CH₃I capture capacities at elevated temperatures. The manuscript is well-organized and articulately written, and I particularly appreciate the open discussion of the recognized limitations of their second strategy. The manuscript is acceptable for publication with minor revision. Following are my questions:

1. "In the manuscript, it is mentioned that at 150 °C and 0.01 bar, MFU-Cu(I) can fully restore its CH₃I uptake capacity after regeneration through washing with polar solvents, as shown in Supplementary Fig. 12. Additionally, the structure remains intact, confirmed by PXRD (Fig. 5e). To strengthen this claim, I recommend including X-ray Absorption Spectroscopy (XAS) results to further validate the structural integrity. This would be especially insightful when compared with the observations of structural damage under conditions of high CH₃I partial pressure."
2. "Regarding Supplementary Fig. 2, the breakthrough uptakes of CH₃I in MFU-Zn-X exhibit the trend OH > Cl > SCN, while the static adsorption test shows the trend Cl > OH > SCN. What could be the reason for this discrepancy? Could it be related to differences in pore size or pore volume?"
3. "The manuscript states that the calculation method adopts the exchange-correlation functional GGA/HCTH based on density functional theory. However, I believe the correct functional should be GGA/HCTH."
4. "It's noteworthy that the CH₃I uptake of 0.41 g/g, as mentioned, closely matches the theoretical value of 0.43 g/g, which is calculated based on a one-to-one correspondence between the counter anion and the CH₃I molecule. How was the theoretical value of 0.43 g/g derived?"
5. "Why was Cu(I) chosen for cation exchange with Zn in MFU-Zn-Cl? According to Supplementary Fig. 12, this leads to a significant reduction (approximately 43.5%) in uptake capacity. Would the performance be better maintained if a different metal was used for the exchange?"

Reviewer #3 (Remarks to the Author):

The capture of radioactive iodine species (namely I₂ & CH₃I) remains a major issue whatever the studied context (nuclear reactor, spent fuel processing units, medical purpose...). MOF materials can constitute an interesting candidate for the removal of these species thanks to their very large specific surfaces areas as well as tunable chemical and structural properties.

Within this frame, authors proposed two innovative strategies for MOF functionalization with the aim to enhance methyl iodide retention at high temperatures and at low concentrations. High adsorption capacities at breakthrough were achieved in challenging conditions (T=150°C, low CH₃I concentration, presence of moisture). In addition, the proposed materials can be regenerated more easily than other industrial adsorbents.

To sum-up, this paper is concise, well-organized and resulted from a great effort to manage with valuable and numerous experimental techniques (synthesis, adsorption, characterization before and after test). I am suggesting that this contribution can be accepted after the following "minor revisions":

Introduction section:

L49: The regeneration of adsorbents is interesting industrially especially for the trapping of volatile iodine species in the spent fuel reprocessing units. However, an irreversible capture is preferred during a nuclear severe accident (as Fukushima) for a nuclear reactor. In that respect, authors are invited to better describe the context of this study. The reference below would be useful: *Annals of Nuclear Energy* 116 (2018) 42–56.

L53: The mechanism of CH₃I adsorption in the presence of silver should be detailed. In addition, it is worth recalling that dispersed silver species (Ag⁺ or charged/metallic clusters) within zeolites are more efficient for methyl iodide adsorption according to these two studies in particular:

Chemical Engineering Journal 379 (2020) 122308 and *Nanomaterials* 2021, 11, 1300.

<https://doi.org/10.3390/nano11051300>.

Results and Discussion:

L120 and L124: authors are invited to indicate also the employed CH₃I concentrations in ppbv or mol/m³ in addition to the partial pressure.

L125-L126: authors are invited to precise the experimental conditions (linear velocity, residence time, adsorbent mass and granulometry and carrier gas (air or inert atmosphere?)) as well as the employed technique for online CH₃I measurements in low concentrations.

L128 and L133: Please add the selected criterion for CH₃I breakthrough for the expression of uptake CH₃I capacities.

L138: the minimal differences at room temperature may arise from quasi-similar specific surfaces areas. I am wondering if authors can provide additional information about the textural properties of the considered adsorbents (specific surface areas, microporous volume,...)

L146: what's the formula of HMTA molecule? Authors are invited to detail the quantity of active sites for the considered adsorbents (Ag⁰, HMTA...). It will be interesting to add an additional graph presenting the global evolution of CH₃I adsorption capacity as function of the number of involved active sites (mmol/g) for the tested supports.

L157: what kind of metallic centers for the considered ECUT-300-200-Ac MOF ?

L222: what's the reason of the decrease of the capacity after the regeneration? decrease of SCN⁻ content or decrease in specific surface areas?

L228: I am wondering about the stability of Cu(I) species? the tests were performed in inert atmosphere? what are the precautions taken by authors to avoid the oxidation of Cu(I) to Cu(II), which can induce parasite oxidation of CH₃I to I₂ ?

Authors are invited to explicit the involved mechanisms (presentation of the equation) for the two proposed strategies of synthesis. Attention should be paid specifically to the nature of interactions between Cu(I) sites and CH₃I where the formation of CuI precipitate is expected to occur.

However, other types of interactions (transfer charge?) are also present according to the authors.

Reviewer #4 (Remarks to the Author):

This manuscript describes two strategies for methyl iodide adsorption using a unified azolate-based metal-organic framework, MFU-4l. The authors performed many and meaningful experiments and provided a lot of convincing information. I have some comments given below.

1. In this study, the authors prepared some adsorbents by themselves for CH₃I capture, e.g., all-silica zeolite H1SL, nitrogen-rich covalent organic framework COF-TAPT, amine-modified metal-organic framework MIL-101-Cr-HMTA, and Ag-containing zeolite Ag⁰-MOR, and compared their adsorption performance. However, there are no characterization data to show if the synthesis results are reasonable or not. So, I am concerned about whether this is an appropriate comparison. I think it would be more reliable to show more than one characterization results to prove that the synthesis was reliable.

2. In Fig. 2b, the exchange of Cl⁻ to OH⁻ or SCN⁻ was indicated by a decrease in the peak for Cl⁻ in XPS. The authors need to specify the binding energy for Cl⁻ precisely, and add reference.

3. ¹H NMR showed that OH⁻ and SCN⁻ were successfully integrated. The authors need to describe exactly what the peak at ~4.7 ppm in ¹H NMR represents.

4. In Supplementary Fig. 7b, the authors need to measure the EDX of the CH₃I-adsorbed MFU-Zn-SCN sample before washing, and subsequently compare the results with the EDX of the regenerated MFU-Zn-SCN to demonstrate the successful regeneration.

5. The binding energy values of the peak mentioned in lines 235-237 are different from the values noted in Fig. S8.

6. There are some minor syntax errors as noted below:

1) The reference format is not consistent.

2) The subscript was not applied properly in the title of the paper written in the reference.

RESPONSE TO REVIEWERS' COMMENTS

Reviewer #1:

In this manuscript, the authors present two innovative sorption methods for chemical capture, both of which can be implemented using a single MOF-based material, MFU-4l. Impressively, these methods show remarkable efficacy in capturing CH₃I at 150 °C, a challenging temperature that is rarely addressed in existing literature. The manuscript successfully demonstrates the utilization of MFU-4l's exchangeable counter anions and cations through two distinct strategies, clearly elucidating the mechanism behind their exceptionally high CH₃I capture capacities at elevated temperatures. The manuscript is well-organized and articulately written, and I particularly appreciate the open discussion of the recognized limitations of their second strategy. The manuscript is acceptable for publication with minor revision. Following are my questions.

Response: We are grateful to the reviewer for their appreciation of our work.

Comments 1: In the manuscript, it is mentioned that at 150 °C and 0.01 bar, MFU-Cu(I) can fully restore its CH₃I uptake capacity after regeneration through washing with polar solvents, as shown in Supplementary Fig. 12. Additionally, the structure remains intact, confirmed by PXRD (Fig. 5e). To strengthen this claim, I recommend including X-ray Absorption Spectroscopy (XAS) results to further validate the structural integrity. This would be especially insightful when compared with the observations of structural damage under conditions of high CH₃I partial pressure.

Response: We thank the reviewer for this insightful comment. Following the reviewer's suggestion, we have included the XAS data for MFU-Cu(I) post adsorption of CH₃I at 0.01 bar (CH₃I@MFU-Cu-0.01bar) in Fig. 5f of the revised manuscript. Supplementary Table 5 and Fig. S10c present the fitting results. The results indicate coordination numbers of 0.8 for Cu-I and 3.1 for Cu-N, suggesting that, on average, each Cu center is approximately bonded to one I atom and three N atoms. This observation aligns with the results from PXRD analysis, verifying the structural integrity following the adsorption of CH₃I at 0.01 bar.

Comments 2: Regarding Supplementary Fig. 2, the breakthrough uptakes of CH₃I in MFU-Zn-X exhibit the trend OH > Cl > SCN, while the static adsorption test shows the trend Cl > OH > SCN. What could be the reason for this discrepancy? Could it be related to differences in pore size or pore volume?

Response: We thank the reviewer for raising this question. In this study, we opted not to conduct static tests; instead, all evaluations were carried out under dynamic conditions. Supplementary

Fig. 2a displays the initial breakthrough curves, and Supplementary Fig. 2b represents the same dataset in a histogram format. There is a consistent correlation between the two presentations.

Comments 3: The manuscript states that the calculation method adopts the exchange-correlation functional GGA/HCTH based on density functional theory. However, I believe the correct functional should be GGA/HCTH.

Response: We thank the reviewer for pointing out this typo. It has been corrected in the revised manuscript.

Comments 4: “It's noteworthy that the CH₃I uptake of 0.41 g/g, as mentioned, closely matches the theoretical value of 0.43 g/g, which is calculated based on a one-to-one correspondence between the counter anion and the CH₃I molecule.” How was the theoretical value of 0.43 g/g derived?

Response: The theoretical value of 0.43 g/g is calculated based on the density of the counter anion SCN and the molecular weight of CH₃I. As indicated in Supplementary Table 1, the density of SCN is measured at 3.06 mmol/g. Multiplying this density by the molecular weight of CH₃I, which is 142 g/mol, yields the theoretical value of 0.43 g/g.

Comments 5: Why was Cu(I) chosen for cation exchange with Zn in MFU-Zn-Cl? According to Supplementary Fig. 12, this leads to a significant reduction (approximately 43.5%) in uptake capacity. Would the performance be better maintained if a different metal was used for the exchange?

Response: We are grateful for the reviewer's insightful comment. The substitution of Cu(I) for Zn(II) within the MFU-4l framework creates open metal sites, which is crucial for enhancing the CH₃I uptake under conditions of elevated temperature and low concentration. The exchange of Zn with other metals does not readily result in the formation of open metal sites.

Following the adsorption of CH₃I at 150 °C and 0.2 bar, MFU-Cu(I) is unable to fully recover its CH₃I adsorption capacity due to a loss of crystallinity. While the hard-soft acid-base theory suggests that the bond between Cu(I) (a soft acid) and N (a soft base) should be strong, this bond still breaks under the specified conditions. This finding raises concerns about the feasibility of enhancing performance stability by substituting with a different metal, especially considering that an alternative metal might significantly reduce the initial CH₃I adsorption capacity.

Reviewer #3:

The capture of radioactive iodine species (namely I_2 & CH_3I) remains a major issue whatever the studied context (nuclear reactor, spent fuel processing units, medical purpose...). MOF materials can constitute an interesting candidate for the removal of these species thanks to their very large specific surfaces areas as well as tunable chemical and structural properties.

Within this frame, authors proposed two innovative strategies for MOF functionalization with the aim to enhance methyl iodide retention at high temperatures and at low concentrations. High adsorption capacities at breakthrough were achieved in challenging conditions ($T=150^\circ\text{C}$, low CH_3I concentration, presence of moisture). In addition, the proposed materials can be regenerated more easily than other industrial adsorbents.

To sum-up, this paper is concise, well-organized and resulted from a great effort to manage with valuable and numerous experimental techniques (synthesis, adsorption, characterization before and after test). I am suggesting that this contribution can be accepted after the following “minor revisions”.

Response: We express our sincere gratitude to the reviewer for dedicating their time to assess our work and offering valuable comments.

Introduction section:

Comments 1: L49: The regeneration of adsorbents is interesting industrially especially for the trapping of volatile iodine species in the spent fuel reprocessing units. However, an irreversible capture is preferred during a nuclear severe accident (as Fukushima) for a nuclear reactor. In that respect, authors are invited to better describe the context of this study. The reference below would be useful: *Annals of Nuclear Energy* 116 (2018) 42–56.

Response: We agree with the reviewer that an irreversible capture is preferred during severe nuclear accidents in order to form a stable product and prevent the re-release of iodine. Following the reviewer’s suggestion, we have revised the introduction to elucidate the context of our research more clearly and have cited the suggested reference.

“...although irreversible capture is preferred in extreme situations like nuclear accidents to prevent the re-emission of radioactive materials, adsorbents with facile regenerability hold significant value in most other scenarios...”

Comments 2: L53: The mechanism of CH₃I adsorption in the presence of silver should be detailed. In addition, it is worth recalling that dispersed silver species (Ag⁺ or charged/metallic clusters) within zeolites are more efficient for methyl iodide adsorption according to these two studies in particular: Chemical Engineering Journal 379 (2020) 122308 and Nanomaterials 2021, 11, 1300. <https://doi.org/10.3390/nano11051300>.

Response: Following the reviewer's suggestion, we have detailed the mechanism of CH₃I adsorption on silver in the revised manuscript based on the suggested references.

“...The adsorption capacity of Ag zeolites for CH₃I mainly depends on the quantity of dispersed Ag species (e.g., Ag⁺ ions or charged/metallic clusters), which, in turn, is governed by the zeolite's cation exchange capacity (CEC). When the quantity of introduced Ag surpasses the CEC, Ag particles begin to develop. These particles exhibit a restricted CH₃I adsorption capacity as only surface atoms are involved in the dissociative chemisorption process (Fig. 1a-ii).”

Results and Discussion:

Comments 3: L120 and L124: authors are invited to indicate also the employed CH₃I concentrations in ppbv or mol/m³ in addition to the partial pressure.

Response: The CH₃I partial pressures used in our study, 0.2 bar and 0.01 bar, are equivalent to 200,000 ppmv and 10,000 ppmv, respectively. These specific values have now been included in the revised manuscript, following the reviewer's suggestion.

Comments 4: L125-L126: authors are invited to precise the experimental conditions (linear velocity, residence time, adsorbent mass and granulometry and carrier gas (air or inert atmosphere?)) as well as the employed technique for online CH₃I measurements in low concentrations.

Response: Following the reviewer's suggestion, we have provided more detailed experimental conditions in the "Dynamic Methyl Iodide Adsorption Experiments" section in the Supporting Information.

“...To test the performance under a partial pressure of 0.2 bar, a nitrogen flow at a rate of 3 mL/min was bubbled through a CH₃I vapor generator and then passed through the adsorbent column with the linear velocity of 18 cm/min and the residence time of about 5 s. CH₃I in the effluent was monitored using an online MS system...To test the performance under a partial pressure of 0.01 bar, a nitrogen flow at a rate of 0.3 mL/min was bubbled through the CH₃I vapor generator and then mixed with another dry nitrogen flow at a rate of 5.7 mL/min before flowing

through the adsorbent column, the linear velocity under this condition is 36 cm/min and the residence time is about 2.5 s...CH₃I in the effluent was monitored using an online MS system...”

Comments 5: L128 and L133: Please add the selected criterion for CH₃I breakthrough for the expression of uptake CH₃I capacities.

Response: We thank the reviewer for this suggestion. The criterion for CH₃I breakthrough is $C/C_0 = 1\%$, meaning that the breakthrough point is reached when the concentration of CH₃I in the effluent stream (C) reaches 1% of the initial concentration (C₀). We have added this criterion in the revised manuscript.

Comments 6: L138: the minimal differences at room temperature may arise from quasi-similar specific surfaces areas. I am wondering if authors can provide additional information about the textural properties of the considered adsorbents (specific surface areas, microporous volume,...)

Response: We appreciate the reviewer’s insightful comments. Following the suggestion, we tested the N₂ adsorption isotherms of MFU-Zn-X (X = Cl, OH, SCN) (**Fig. R1**). The results revealed a notable reduction in the specific surface area and pore volume after anion exchange, suggesting a partial destruction of the crystalline structure. Therefore, the minimal discrepancies observed in CH₃I uptakes at room temperature should not be attributed to quasi-similar specific surface areas.

As elucidated in our prior publication (*Nat. Commun.* **2022**, *13*, 2878), at room temperature, the adsorption of CH₃I is primarily determined by the number of binding sites. We believe that, in the present study, the comparable densities of counter anions in MFU-Zn-X variants (Cl⁻: 3.25 mmol/g; OH⁻: 3.34 mmol/g; SCN⁻: 3.06 mmol/g) account for their minimal differences in CH₃I uptake at room temperature.

Fig. R1 N₂ adsorption isotherms of MFU-Zn-X (X = Cl, OH and SCN) collected at 77 K.

Comments 7: L146: what's the formula of HMTA molecule? Authors are invited to detail the quantity of active sites for the considered adsorbents (Ag^0 , HMTA...). It will be interesting to add an additional graph presenting the global evolution of CH_3I adsorption capacity as function of the number of involved active sites (mmol/g) for the tested supports.

Response: The formula of HMTA (hexamethylenetetramine) is $\text{C}_6\text{H}_{12}\text{N}_4$. Following the reviewer's valuable suggestion, we have detailed the quantity of active sites for the evaluated adsorbents in Supplementary Table 1. Furthermore, we have included a figure depicting the CH_3I adsorption capacity at 0.01 bar and 150 °C in relation to the number of presumed adsorptive sites across various tested adsorbents (Supplementary Fig. 5 of the revised manuscript).

As illustrated in this figure, among all tested adsorbents, only Ag^0 -MOR and MFU-Zn-SCN exhibit CH_3I uptakes that closely approach theoretical values. These results further reinforce the notion that, under high-temperature and low-concentration conditions, the equilibrium adsorption capacity is exclusively determined by the number of strong chemisorption sites, while high reactivity of the adsorption site is essential for achieving this capacity.

It should be mentioned that the high I/Ag ratio in Ag^0 -MOR may result from its low Si/Al ratio (6.5) and low Ag content (6.3 wt%), which favors the formation of dispersed Ag species to enhance CH_3I uptake.

Comments 8: L157: what kind of metallic centers for the considered ECUT-300-200-Ac MOF?

Response: The metallic centers of ECUT-300-200-Ac are Cd and U. This information has been included in the revised manuscript.

Comments 9: L222: what's the reason of the decrease of the capacity after the regeneration? decrease of SCN^- content or decrease in specific surface areas?

Response: We thank the reviewer for raising this point. In response, we compared the original MFU-Zn-SCN with the material after undergoing two adsorption-regeneration cycles (re-MFU-Zn-SCN) through characterizations. N_2 adsorption isotherms revealed a moderate reduction in specific surface area after regeneration (**Fig. R2a**). Spectroscopic examinations via ATR-FTIR (**Fig. R2b**) and XPS (**Fig. R2c**) showed a notable decrease in the SCN^- content of the re-MFU-Zn-SCN compared to the pristine material. As discussed earlier, we have demonstrated that the specific surface area has a minimal impact on CH_3I uptake under conditions of high temperature and low CH_3I concentration. Therefore, the decreased CH_3I adsorption capacity can be primarily attributed to the lower SCN^- content in the regenerated adsorbent. However, it is possible that the

reduction in SCN^- content is indirectly linked to the decreased surface area, which hampers the complete restoration of SCN^- sites through ion exchange.

Fig. R2 N_2 adsorption isotherms collected at 77 K (a), ATR-FTIR spectra (b) and S 2p XPS spectra (c) of MFU-Zn-SCN before and after two adsorption-regeneration cycles (designated as re-MFU-Zn-SCN).

Comments 10: L228: I am wondering about the stability of Cu(I) species? the tests were performed in inert atmosphere? what are the precautions taken by authors to avoid the oxidation of Cu(I) to Cu(II), which can induce parasite oxidation of CH_3I to I_2 ?

Response: We thank the reviewer for raising this point. Indeed, exposing Cu(I) to the air environment can lead to a conversion of a portion of Cu(I) into Cu(II), resulting in a change in sample color from beige to green. This observation is supported by our EPR and XANES results conducted in an air environment, as illustrated in Supplementary Fig. 9b and c. However, this valence transformation is reversible. Following high-temperature inert gas activation or vacuum treatment, the sample color reverts from green to beige, and Cu(II) is converted back into Cu(I). This reversion is confirmed by our XPS results obtained under vacuum conditions, as demonstrated in Supplementary Fig. 9a.

All the tests conducted in this study were performed in a N_2 atmosphere. Prior to the tests, the samples were activated under high-temperature inert gas conditions to maintain the Cu(I) state.

Comments 11: Authors are invited to explicit the involved mechanisms (presentation of the equation) for the two proposed strategies of synthesis. Attention should be paid specifically to the nature of interactions between Cu(I) sites and CH_3I where the formation of CuI precipitate is expected to occur. However, other types of interactions (transfer charge?) are also present according to the authors.

Response: The first strategy employs a metathesis reaction, where the counter anion X^- ($X = \text{Cl}, \text{OH}, \text{ or } \text{SCN}$) present in MFU-Zn- X acts as a nucleophile. It attacks the C-terminal of CH_3I , leading to the dissociation of the C-I bond. The formed I^- replaces the X^- to act as the counter anion for MFU-Zn, while the generated CH_3X is expelled from the adsorbent by the carrier gas. The whole process can be presented by the following equation:

The second proposed strategy is based on non-dissociative chemisorption. Instead of forming CuI , as suggested by the reviewer, the open Cu(I) sites can strongly interact with the iodine end of CH_3I *via* non-dissociative chemisorption as presented by the following equation:

The EPR and XANES spectra of CH_3I loaded MFU-Cu(I) ($\text{CH}_3\text{I}@\text{MFU-Cu(I)}$), as depicted in Figure 5c and d, verify that the open Cu(I) sites maintain their monovalent state after interaction with CH_3I . This observation suggests a charge transfer from the electron-rich segment of CH_3I , specifically from the iodine end, to the open Cu(I) site.

We sincerely thank the reviewer for this comment. Please kindly note that the mechanisms underlying the two strategies are illustrated in the schematic diagrams provided in Fig. 1b.

Reviewer #4:

This manuscript describes two strategies for methyl iodide adsorption using a unified azolate-based metal-organic framework, MFU-4l. The authors performed many and meaningful experiments and provided a lot of convincing information. I have some comments given below.

Response: We sincerely thank the reviewer for their encouraging comments.

Comments 1: In this study, the authors prepared some adsorbents by themselves for CH₃I capture, e.g., all-silica zeolite HISL, nitrogen-rich covalent organic framework COF-TAPT, amine-modified metal-organic framework MIL-101-Cr-HMTA, and Ag-containing zeolite Ag⁰-MOR, and compared their adsorption performance. However, there are no characterization data to show if the synthesis results are reasonable or not. So, I am concerned about whether this is an appropriate comparison. I think it would be more reliable to show more than one characterization results to prove that the synthesis was reliable.

Response: We thank the reviewer for raising this point. The benchmark adsorbents, COF-TAPT and MIL-101-Cr-HMTA, were provided by our former group members, Yaqiang Xie and Baiyan Li, who have authored the original studies published in *Nat. Commun.* **2022**, *13*, 2878, and *Nat. Commun.* **2017**, *8*, 485, respectively. The other two benchmark adsorbents, HISL and Ag⁰-MOR, were synthesized in accordance with the methods described in the literature. The PXRD patterns for HISL and Ag⁰-MOR validate their respective phase-pure crystalline structures, as illustrated in **Fig. R3a**. The Ag content in Ag⁰-MOR is determined using ICP-OES to be 6.3 wt%. The strong hydrophobicity of HISL is confirmed by the minimal water uptake observed in the water adsorption isotherm (see **Fig. R3b**).

Fig. R3 (a) PXRD patterns of HISL and Ag⁰-MOR. (b) Water adsorption isotherm of HISL collected at 298 K.

Comments 2: In Fig. 2b, the exchange of Cl^- to OH^- or SCN^- was indicated by a decrease in the peak for Cl^- in XPS. The authors need to specify the binding energy for Cl^- precisely, and add reference.

Response: We thank the reviewer for this suggestion. Based on the literature (Chem **2018**, *4*, 2894), the binding energy for $\text{Cl} 2p$ is approximately 198.3 eV. We have included this value in the revised manuscript and cited the reference accordingly.

Comments 3: ^1H NMR showed that OH^- and SCN^- were successfully integrated. The authors need to describe exactly what the peak at ~4.7 ppm in ^1H NMR represents.

Response: The peak at ~4.7 ppm in ^1H NMR represents the protons of the hydroxyl groups (*Solid State Nuclear Magnetic Resonance* **2022**, *119*, 101793). Following the reviewer's suggestion, we have made this clear in the revised manuscript.

Comments 4: In Supplementary Fig. 7b, the authors need to measure the EDX of the CH_3I -adsorbed MFU-Zn-SCN sample before washing, and subsequently compare the results with the EDX of the regenerated MFU-Zn-SCN to demonstrate the successful regeneration.

Response: Following the reviewer's suggestion, we conducted a comparative analysis of the EDX results for CH_3I -adsorbed MFU-Zn-SCN and its regenerated counterpart, presented in Supplementary Fig. 8b of the revised manuscript. The simultaneous emergence of the S peak alongside the vanishing I peak indicates the effective regeneration of the adsorbent.

Comments 5: The binding energy values of the peak mentioned in lines 235-237 are different from the values noted in Fig. S8.

Response: We have corrected this error in the revised manuscript.

Comments 6: There are some minor syntax errors as noted below:

- 1) The reference format is not consistent.
- 2) The subscript was not applied properly in the title of the paper written in the reference.

Response: We thank the reviewer for pointing out these errors. We have corrected them in the revised manuscript.

REVIEWERS' COMMENTS

Reviewer #1 (Remarks to the Author):

The manuscript is well revised and can be accepted.

Reviewer #3 (Remarks to the Author):

I would like to thank the authors for their investment to consider almost all the suggested remarks. The revised draft of the article can be accepted directly for publication.

Reviewer #4 (Remarks to the Author):

I confirmed that all of my comments were applied well. So, I recommend to publish the manuscript as is.